# Solubility, Rheology, and Coagulation Kinetics of Poly-(O-Aminophenylene)Naphthoylenimide Solutions

**DOI:** 10.3390/polym12112454

**Published:** 2020-10-23

**Authors:** Ivan. Y. Skvortsov, Valery G. Kulichikhin, Igor I. Ponomarev, Lydia A. Varfolomeeva, Mikhail S. Kuzin, Kirill M. Skupov, Yulia. A. Volkova, Dmitry Y. Razorenov, Olga A. Serenko

**Affiliations:** 1A.V. Topichiev Institute of Petrochemical Synthesis RAS, 29 Leninsky Av., 119991 Moscow, Russia; klch@ips.ac.ru (V.G.K.); varfolomeeva.lidia@mail.ru (L.A.V.); gevahka15@gmail.com (M.S.K.); 2A.N. Nesmeyanov Institute of Organoelement Compounds of Russian Academy of Sciences, 28 Vavilova St., 119991 Moscow, Russia; gagapon@ineos.ac.ru (I.I.P.); yvolk@ineos.ac.ru (Y.A.V.); razar@ineos.ac.ru (D.Y.R.); o_serenko@ineos.ac.ru (O.A.S.)

**Keywords:** rheology, coagulation, heat resistance fibers, heterocyclic polymer, polynaphthoylenebenzimidazole (PNBI), ladder polymers, poly-(o-aminophenylene)naphthoylenimide precursor, spinning solution, solution modification, polymer solution rheology

## Abstract

The effect of temperature and storage time at a constant temperature on the stability of poly-(o-aminophenylene)naphthoylenimide solutions in N-methylpyrrolidone has been analyzed using rotational rheometry. A temperature–time window beyond which an irreversible change in the viscoelastic properties of solutions due to cumulative reactions of continuous polymerization and possible intramolecular cyclization has been detected. The influence of polymer concentration and its molecular weight on the rheological properties of solutions determining the choice of methods for their processing into fibers and films has been investigated. The effect of non-solvents (water and ethanol) additives on the rheological properties of solutions and the kinetics of their coagulation has been studied. Dosed addition of non-solvent into the solution promotes a significant increase in the viscoelasticity up to gelation and phase separation. Non-solvent presence in the polymer solutions reduces the activity of the solvent, accelerates the movement of the diffusion front at coagulation, and minimizes the number of macro defects. The combination of parameters under investigation renders it possible for the first time to develop new principles modifying dopes for wet spinning into aqueous or ethanol coagulation bath and finally to obtain a heat- and fire-resistant polynaphthoylenebenzimidazole fibers.

## 1. Introduction

Fibers made from polymers with the high energy of chemical bonds (significantly higher than that of flexible-chain polymers) are heat-resistant and capable to preserve functional properties in the air up to temperatures exceeding 200–250 °C and reaching 300–350 °C and even higher for certain types, that provides their long-term use [1,2,3]. This is usually achieved by introducing heteroatoms such as N, S, and P into the polymer chain. Being a part of condensed heterocycles, these elements increase the kinetic and equilibrium rigidity of macromolecules. Several years ago, the scientific group of the A.N. Nesmeyanov Institute of Organoelement Compounds of Russian Academy of Sciences developed a technology for the production of a ladder polynaphthoylenebenzimidazole (PNBI) soluble in sulfuric and methanesulfonic acid [4]. Fibers, spun from these solutions, were used as heat-shielding felt on some aerospace objects.

In the current paper, the synthesis of the PNBI family is based on polycondensation of aromatic tetraamines and 1,4,5,8-naphthalene tetracarboxylic acid dianhydride via the stage of stable prepolymer-poly-(o-aminophenylene)naphthoylenimide (PANI) preparation [4,5,6,7,8], soluble in N-methyl-2-pyrrolidone (NMP). Subsequent heat treatment of fibers and films spun from prepolymers solutions leads to the formation of heat and fire-resistant materials possessing a record retention temperature of functional properties reaching 450 °C, oxygen index up to 95, resistance to radiation, ultraviolet radiation, and extremely high climatic stability [3,6]. The use of PNBI varnishes in high-temperature electrical engineering [7] is also of particular interest.

The ideas of proceeding with all synthetic procedures in the NMP as an organic solvent are developed through the stage of forming a soluble and recyclable prepolymer in contrast to the well-known environmentally unfriendly process for obtaining PNBIs in polyphosphoric and other mineral acids.

The above intro was necessary to clarify the scientific novelty of synthetic procedures allowing us to prepare pre-polymers dissolved in an organic solvent. Earlier, the authors [8] investigated the hydrodynamic properties of PANIs with various bridging groups by sedimentation and viscometry methods. The constants in the Mark–Kuhn–Houwink equation were determined and the possibility of synthesizing high-molecular prepolymer (up to 100 kg/mole) PANI with ether bridge-group (PANI-O) in NMP was proven.

An important feature of PANI-O solutions in NMP is their instability [9] due to the processes of continuous polymerization and possible intramolecular cyclization occurring in time. Besides, it is the possible formation of a physical gel due to hydrogen bonding between free ortho-amino groups and carbonyl groups of naphthoylenimide rings in neighboring polymer chains. The study of PANI-O solutions instability requires a systematic investigation of rheological properties in a wide range of compositions and temperatures. This approach makes it possible to select certain temperature–time conditions for storing solutions or, more precisely, their conditioning for spinning, which implies the achievement of certain viscoelastic properties followed by the selection of the optimal coagulation bath composition, which plays a key role in obtaining defect-free fibers by the wet spinning method.

Wet spinning is a well-studied technological process for producing fibers by extrusion of a polymer solution from a spinneret immersed in a suitable coagulant, where the solution jets transform to as-spun fibers due to mass transfer between the solvent that diffuses from jets and coagulant diffusing into the jets [10]. A lot of publications describe the influence of the nature and thermodynamic parameters of the polymer solution and coagulation composition for wet and dry-wet spinning of fibers [11,12,13,14,15,16]. As an example, the correct choice of the coagulant in the production of heat-resistant polyamide fiber made it possible to enhance its mechanical properties by ~2 times [12]. A detailed selection of a coagulant for spinning fibers from polyacrylonitrile (PAN) solutions [13,14] resulted in a significant reduction of the defectiveness of the fiber. It is important to note that the choice of a solvent-coagulant pair is an extremely critical step in the wet spinning process. When “constructing” such a pair, several key requirements must be taken into account. Firstly, these components must be soluble each in the other, at least in the proportions that are realized during spinning. Secondly, for this pair, the separation conditions in the processes of solvent and coagulant regeneration and recovery should be identified. Thirdly, the optimal composition of the pair should be of a certain activity towards interdiffusion interaction during fiber spinning. Such activity ensures smooth phase decomposition of the polymer solution when fibers without defects are being formed [15].

Another strategy is to add a certain proportion of a non-solvent into the polymer solution to control the rheological properties and, what is the most important, to change the parameters of polymer–solvent compatibility. For example, the addition of water, which is a hard precipitant, to solutions of polyacrylonitrile (PAN) in dimethyl sulfoxide (DMSO) made it possible to reduce significantly the number of finger-like macropores in the fiber and level the difference in the transverse morphology of the core-shell formed during the wet spinning [17]. Following this logic, an important part of this work was to study the effect of the addition of coagulants such as water and ethanol into the spinning solutions on the rheological properties and the kinetics of coagulation of PANI-O solutions. Thus, one of the main tasks of the work is to reveal the influence of the rheological parameters of the solution varying in time, on the one hand, and non-solvent additions to the solution effect on the rheological properties and coagulation processes of PANI-O spinning solutions, on the other. This approach is of fundamental importance in developing a strategy for the spinning process of insufficiently explored polymer solutions.

## 2. Materials and Methods

### 2.1. Materials

#### 2.1.1. Monomers and Solvents

3,3′,4,4′-tetraaminodiphenyl ether (Rubezhnoe plant, Rubezhnoe, Ukraine) with a melting point of 155–156 °C and 1,4,5,8-naphthalenetetracarboxylic acid dianhydride (VNIPIM, Tula, Russia) were dried at 100 °C under vacuum before use. Benzimidazole, benzoic acid, and NMP (Thermo Fisher Scientific, Acros Organics, Waltham, MA, USA) were used as received.

In general, the scheme for obtaining a heat-resistant, non-combustible PNBI fiber is shown in Figure 1. It consists of two stages: The synthesis of the PANI-O prepolymer and its transformation in PNBI at temperatures of 250–350 °C due to cyclization with the removal of water:

#### 2.1.2. Synthesis of PANI-O

2.3027 g (0.01 mol) of 3,3′,4,4′-tetraaminodiphenyl ether, 2.6818 g (0.01 mol) of 1,4,5,8-naphthalenetetracarboxylic acid dianhydride, 0.24 g (0.00196 mol) of benzoic acid, 0.24 g (0.002 mol) of benzimidazole, and 20 mL of anhydrous NMP were placed in the flask and stirred in Ar flow for 24 h at room temperature forming a viscous solution of the polymer. The reaction mixture was then stirred at 50–60 °C until the agitator was almost stopped due to very high viscosity. This method was used to obtain polymer samples with an intrinsic viscosity of 0.6–1.2 dL/g and solution concentrations from 10% to 15% (mass). Further growth of the molecular weight of the samples was controlled by isothermal heating of the solutions in the operating unit of the rotary rheometer at a chosen temperature to achieve the complex of viscoelastic properties (due to the increase of the molecular weight), necessary for successful fibers spinning.

#### 2.1.3. PANI-O Solutions

Solutions of the prepolymer with different molecular weights were obtained by controlled heating at 70 °C by stirring with a J-shaped stirrer rotating at 10 rpm. The intrinsic viscosity values of pre-polymer during controlled heating were determined by standard dilution and subsequent extrapolation of η_sp_/C values towards zero concentration. To estimate the effect of polymer concentration on the coagulation process and rheological properties of solutions, a series of solutions with concentrations from 1 to 23% were prepared based on a 12% reaction solution P1 with [η] = 1.5 dL/g (see Table 1). Dilution with NMP was carried out by stirring for 3 h at 25 °C, while an increase of the concentration was reached in a vacuum rotary evaporator at 25 °C at a residual pressure of 5 × 10^−5^ bar. To estimate the influence of molecular weight, assessed by the change of the intrinsic viscosity, the 10% PANI-O solution was heated up to get the intrinsic viscosity values equal to 0.6, 1.2, and 1.8 dL/g (determined in Ubbelohde viscometer at 25 °C by ASTM D2857 [18]). In this way solutions P2, P3, and P4 (Table 1) were prepared.

Samples of solutions containing different amounts of water or ethyl alcohol were obtained by their adding at continuous stirring in sealed vials at a stirrer speed of 100 rpm. To study the effect of the non-solvent content on the rheological properties and stability of solutions a series of PANI-O solutions with water (PH) and ethanol (PE) were prepared. For these systems, the weight fraction of non-solvent was calculated in terms of the fraction of NMP keeping constant the ratio of PANI-O to NMP. Each sample was stirred for 2 h before the estimation of the non-solvent effect on rheological behavior, the morphology of the solution, and features of the solution droplet coagulation process under influence of coagulant.

The relevant information on compositions of the series of solutions under consideration and values of intrinsic viscosity of the polymer are given in Table 1. M_υ_ values were calculated using the Mark–Kuhn–Houwink Equation in accordance with the parameters determined in [8].

### 2.2. Methods

#### 2.2.1. Modeling of Fiber Spinning

The morphology evolution of solution droplet (modeling the jet/fiber cross-section) surrounded by a coagulant was studied following a previously developed method [14]. The thickness of the droplet was ~0.1 mm and its average diameter ~1.5 mm. The observation of polymer solution-coagulant interaction was carried out using the Biomed 6PO microscope (Biomed Co, Moscow, Russia). Tests for all coagulants were performed at 25 °C.

Water (distilled, conductivity < 1.0 µS/cm) and ethanol (Ecos-1, Moscow, Russia) were tested for modeling of the wet spinning process.

#### 2.2.2. Rheology

The rheological properties of solutions were measured in steady-state and oscillation regimes of the shear strain using rotational rheometer Anton Paar MCR 301 in the temperature range of −17–70 °C. The behavior of “living” solutions was tested using an operation unit of two geometries. Coaxial cylinders with a diameter of the inner cylinder of 10 mm and a gap of 0.42 mm were used for the prolong kinetics measurements, cone and plate unit with a cone diameter of 25 mm and an angle between cone and plate of 2° being used for the tests at different shear rates and temperatures. 

Kinetic measurements were performed at non-destructing mechanical action in the cylinder-cylinder unit at a constant frequency of 1 Hz and a strain of 1%. The edge of the gap was covered by silicone oil to avoid gel formation due to solution interactions with wet air. The flow curves were recorded at the shear rate range of 10^−2^–10^4^ s^−1^, the amplitude dependences of the complex elasticity modulus were recorded at the strain range of 0.06–62 rad/s at two frequencies: 1 and 80 Hz, respectively. The frequency dependences of the storage and loss moduli were measured in the angular frequency range of 0.6–628 rad/s. 

The scanning of frequency was performed every 20 min at a constant temperature to determine the frequency dependences of moduli at the strain of 1%. In this way, the changing intensity of viscoelastic characteristics at constant temperature was estimated. These data allow choosing the heating time for further dopes conditioning.

## 3. Results and Discussion

### 3.1. Rheology

Rheological parameters and even solubility of PANI-O depends on chemical and physicochemical transformations in the solutions. The main chemical processes are an increase in the molecular weight of the polymer, heterocyclization at > C=O and –NH_2_ groups, and formation of a physical network in concentrated solutions. Their kinetics is determined by the temperature and concentration of the polymer in the solution and by the selected solvent. The processes of cyclization and molecular weight growth are most active during synthesis at elevated temperatures. In the resulting polymer solution, the processes slow down but do not stop. The solution is still “alive” and significantly changes its properties depending on the duration and temperature of storage. Consequently, post-polymerization conditioning is needed to attain the rheological properties of solutions to the range most convenient for fibers spinning.

The “freezing” of chemical and physicochemical processes in the reaction solution immediately after the synthesis is the most advanced technique for its further controlled conditioning. So, the protocol of solution preparation for testing was the following: Storage at low temperatures, heating to the required temperature, and conditioning until reaching the appropriate rheological properties for successful fiber spinning.

Data on the influence of low temperatures on rheological properties for P1 solution are presented in Figure 2.

The solution becomes more structured when it cooled to −17 °C. Its elasticity begins to increase, while the slopes of the frequency dependences G’ and G’’ decrease to the same value ~0.6. Cooling does not improve the spin-ability of concentrated systems, since the loss and elasticity moduli are becoming the same value causing instability during flow out of the spinneret die. The kinetics of changes in the rheological properties of PANI-O solutions measured at 25 °C after storing at different temperatures (−196, −17, and 25 °C) for a long time were analyzed by the value of the Newtonian viscosity (Figure 3a) and the position and the shape of the flow curves (Figure 3b).

The rheological properties begin to change within 5 days after the synthesis at a storage temperature of 25 °C. In 20 days, the viscosity increases a thousand times and the solution becomes a structured fluid. When cooled to −17 °C, the solution does not cause obvious gelation since the dissipative component of the complex modulus exceeds the elastic one (see Figure 2). However, their proximity may indicate the inhibition of the pre-polymer gelation or cyclization reaction because rheological properties of the solution begin to change only on the 60th day of storage (Figure 3a).

The possibility of rapid freezing of solutions and their storage at a liquid nitrogen temperature (−196 °C) was of particular interest, where the processes of self-diffusion of macromolecules are extremely limited in addition to slowing down the rate of chemical reactions. As shown in preliminary experiments with serial cycles of freezing and melting, the rheological properties of the system remain unchanged, and optical microscopy does not allow revealing any differences between the original and melted samples, which makes this method preferable for long-term storage.

Nevertheless, moderately low storage temperatures of ~−17–−20 °C can be considered as the most preferred ones due to the absence of a difference in the behavior of solutions before and after cooling and significantly faster heating of the sample to a given temperature before the target experiment.

#### 3.1.1. Molecular Weight Effect

The rheological properties of a solution are governed by a polymer concentration and its initial molecular weight, which can change significantly upon heating due to the inducing chemical transformations in the polymer. To determine the required time for the solution “aging”, a preliminary experiment was carried out in the rheometer unit which allows fixing the required heating time, i.e., conditions for obtaining a solution with viscoelastic characteristics suitable for fiber spinning.

Figure 4 shows the frequency sweeps of the components of the complex modulus of elasticity measured at 70 °C after different heating times at 70 °C for a 10% P2 solution.

In a solution heated to 70 °C, significant changes occur. At the beginning of heating, the low-viscous solution is a viscoelastic liquid. The values of the frequency dependence slopes are ~1.9 for G’ and ~1.0 for G’’, respectively. With prolong heating, the slopes become equal, i.e., the degree of structuring increases significantly (see Figure 4b).

To monitor the intrinsic viscosity evolution, the P2 solution was heated for 3 and 5 h, and its values became 1.2 and 1.8 dL/g, respectively. This fact indicates a significant increase in the molecular weight of the sample upon heating and a corresponding increase in the structuring of the solutions when [η] exceeds 1 dL/g.

Thus, controlled heating allows improvement of the viscoelastic properties of dopes over a wide temperature range and indicates the importance of temperature control during spinning.

#### 3.1.2. Concentration Effect

The second parameter that has a significant effect on rheological properties is the concentration of the solution. Experimental data for a series of P1 solutions are presented in Figure 5.

As the concentration increases, the values of the moduli of elasticity and loss goes up, but the difference between them decreases. At the same time, starting with a concentration of 9% and higher, the solutions have close values of frequency dependence slopes: G’ (0.85) and G’’ (0.75), which indicates the growth of solutions structuration. At lower concentrations, the G’(ω) dependence becomes more pronounced in the entire frequency range with an increase in the slope up to 1.3, but it fails to be reliably recorded at concentrations below 6%.

According to the flow curves for solutions of different concentrations (Figure 6), the appearance of a viscosity anomaly for the P1 series starts from a concentration of 9%. As is seen from these data, only when the polymer concentration is above 12–15% the values of the Newtonian viscosity reach 100 Pa·s and at 23% the viscosity approaches 1000 Pa·s. In other words, both strategies, i.e., an increase in the molecular weight of the polymer or concentration of the pre-polymer solution do not allow producing of the dopes with viscoelastic properties suitable for dry-wet or mechanotropic spinning [19,20].

Working with relatively low molecular weight polymers in solutions could allow achieving high values of viscosity when concentration is close to 20%, but the elasticity of the solutions is insufficient to form a stable jet when flowing from the spinneret die. With an increase in molecular weight, the system quickly becomes extremely elastic too fast, and it becomes hard to find the appropriate solution elasticity suitable for stable jet formation. As a result, the most acceptable method for processing PANI-O solutions is the classic wet-spinning method.

The main features of this method consist of interdiffusion processes at the interaction of the thin jets of the dope with a coagulant, i.e., non-solvent. This interaction results in the phase separation of polymer dopes with segregation of the polymer phase. Among popular coagulants of PANI-O solutions in NMP, ethanol and water are considering often. Both of them are soluble in NMP. That is why we solved to introduce these non-solvents to PANI-O solutions to imagine the evolution of rheological properties of as-spun fibers in a transient state from solution to gel, that impossible to understand in the real process of fibers spinning. From this viewpoint, such an approach could be classified as modeling of viscoelastic properties evolution of the dope at contact with coagulants. From the other side, a transformation of inter- and intramolecular interactions in solutions in a presence of non-solvent could create new rheological behavior, as well as the specific thermodynamic and kinetic conditions at interaction with a coagulant. The complex of these events will be considered below.

#### 3.1.3. Ethanol Effect

Ethanol is a coagulant for PANI-O solutions in NMP, while as was shown earlier [9], the addition of an ethanol/NMP mixture in a 30/70 ratio does not lead to the coagulation of a polymer solution which indicates the definite degree of compatibility of ethanol with NMP. To check the compatibility limit of these components a series of PANI-O solutions in NMP containing an increasing amount of ethanol was prepared to determine its critical concentration corresponding to the onset of phase separation. The flow curves and frequency dependences of the moduli of such compositions are shown in Figure 7.

Low ethanol concentrations (up to ~7%) enhance the viscosity anomaly. Moreover, the viscosity increases in the region of low stresses whereas it decreases in the region of high stresses in comparison with the flow curve of the PANI-O solution in NMP. At the same time, a narrowing of the maximal Newtonian viscosity plateau begins to be observed, i.e., the system exhibits “non-Newtonian” properties over a wider range of shear stresses. Starting from ~10% of ethanol, the absolute values of the viscosity at low stresses increase, reaching a centuple increase for an 18% solution compared to the initial solution, and at 20% of ethanol content, a distinct yield point appears on the flow curve.

The following regions can be distinguished for all three-component systems on the dependence of viscosity on shear stress as the stress increases: The maximal Newtonian viscosity, pseudoplastic behavior with different values of exponent in the power-law of flow, viscoplastic behavior with a pronounced yield point, and the transition region corresponding to spurt. It can be assumed that the gradual transition from the plateau with the highest viscosity to the spurt characterizes the successive destruction of the intermolecular interactions, starting from the topological network of entanglements towards the destruction of stronger hydrogen bonds. The realization of these processes under the influence of shear deformations leads to the orientation of polymer chains and the loss of their relaxation mobility that provide a crucial contribution to the adhesion of the solution to the solid surfaces of the operating unit causing the effect of slipping. 

The effect of ethanol on the viscoelastic properties of the systems under investigation is demonstrated in the frequency dependences of the tan(δ) (Figure 8). 

The tan(δ) decreases with increasing frequency for NMP solution, which reflects the basic principle of temperature–time superposition, i.e., the system becomes more rigid not only with decreasing temperature but also with increasing frequency of deformation. The introduction of small amounts of ethanol (up to 7%) reduces the intensity of this dependence, leading to almost complete frequency invariance. The dependence becomes negative at higher ethanol concentrations in the low-frequency range. It “softens” with an increase of the frequency intensity. A crossover point appears with a further increase in the dissipative component at an ethanol content above 13%.

#### 3.1.4. Water Effect

Water is a more active coagulant in comparison with ethanol, causing the phase separation of the PANI-O solution already at 15% content [9]. The frequency dependences of the viscoelastic characteristics of solutions with aqueous additives are shown in Figure 9.

The addition of water structures the solutions, causing an increase in the absolute values of the moduli (while the elastic modulus increases faster, especially in the region of low frequencies), their convergence, and a decrease in the slope angles of the frequency dependences G’and G’’. With the addition of 14% water, a “collapse” of the values of the moduli occurs since the system begins to behave like a viscoelastic solid, i.e., G’ becomes higher than G’’ and ceases to depend on frequency. This gel-like system is characterized by slippage that reflects on a decrease of the absolute values of moduli.

The effect of water on the relaxation properties of ternary systems is similar to the effect of ethanol, but with a shift towards lower concentrations of the coagulant. The corresponding data are presented in Figure 10.

Up to a water concentration of 5–7%, the tangent decreases (apparently, due to an increase in G’) as the strain frequency increases. This effect occurs up to the formation of a heterophase system characterized by an increase in tan(δ) values with the frequency due to a decrease in G’. In the high-frequency region, the G’’/G’ ratio of both ethanol and water-containing systems approaches the values typical for the initial solution in NMP, which indicates an analogy in the change in the relaxation properties of binary and ternary systems. With a further increase in concentration, the qualitative differences begin to appear in the relaxation behavior of aqueous and ethanol systems. Thus, the addition of 14% water to the solution leads to phase separation, while the systems with ethanol remain stable up to an ethanol content of about 20%.

The flow curves of solutions containing different amounts of water are shown in Figure 11.

The water effect is partly different from the ethanol effect under conditions of stationary deformation. Firstly, the structuring of the solution is observed in the entire range of shear rates. As a result, the region of the Newtonian viscosity becomes less visible even at a content of water ~2%. Secondly, an increase in viscosity with an increase in the fraction of introduced water to a concentration of 7%, (which is approximately 70% of the maximum possible content in the homogeneous solution), occurs smoothly without a sharp jump in viscosity. If ethanol presents in the solution, this concentration corresponds to ~30% of the maximum possible.

An increase in water concentration up to 14% leads to phase separation, i.e., a structured heterophase system with a yield point is formed. The rheological data show a slippage of heterophase medium relatively cone and plate of the rheometer. When ethanol is used as a non-solvent additive, such a metamorphosis with the rheological state of the solution is observed only at its content close to 20%. In the concentration range from 2 to 14% the system exhibits a clear non-Newtonian behavior without the region of the constant Newtonian viscosity, which may indicate the destructive change of the system even at low stresses. The flow curves of these systems do not have a change in the slope angle and shape along with water content, while in the case of ethanol additives flow curves are characterized by a z-shape (see Figure 7a).

It could be seen that at the spurt region the viscosity of solutions with the addition of water is reduced to the same order of values, found for systems with ethanol. In the region of critical shear rates, the stress of the spurt beginning, which corresponds to the viscosity η_s_, appears at 2 kPa, while the values of η_s_ are dependent significantly on the nature of the non-solvent introduced (Figure 12).

There is a weak dependence of the viscosity corresponding to spurt onset on ethanol or water content for in NMP-PANI-O solutions. In the case of water adding the value of η_s_ increases almost linearly, which indicates the incomplete destruction of the supramolecular structure of the solution during shear and its greater strength in comparison with the initial solution and a solution with ethanol.

Dependences of the maximal Newtonian viscosity on the concentration of ethanol and water introduced into the PANI-O solution in NMP (Figure 13) have a similar character. Additives only slightly increase the viscosity of the system until a certain concentration, after which the slope changes and the dependences become stronger. In both cases, the system is actively structured due to the loss of NMP affinity to the polymer in the presence of non-solvents. For ethanol, as a softer coagulator, a break is observed at ~7%, and a further increase in viscosity is less distinctive (the slope is 0.23), while these characteristics for water are 5% and 0.6, respectively. This indicates a higher activity of water in structuring the solution, although it does not reflect the strength of the forming in solutions structure.

The above-mentioned experimental data and reasoning about the solutions structuring in the presence of non-solvents require at least a hypothetical understanding of the possible mechanism of structure formation. However, to obtain additional information about the processes occurring during the fibers and films spinning under the action of coagulants, it is advisable to consider the process of coagulation of solution drops simulating the cross-section of the spinning fiber. That is fundamentally important for the selection of wet-spinning regimes.

### 3.2. Coagulation

The liquid jet solidification and the solvent removal kinetics are determined by the interaction of the solution with the coagulant in the wet spinning process [11]. It is common to divide coagulants into “hard” and “soft” according to the activity of interaction with the polymer solution. The use of “hard” coagulants is often not the best choice, since a high diffusion rate can lead to a non-uniformity of the diameter of the resulting fiber, especially in combination with its stretching, the formation of the “shell-core” morphology, and the formation of defects in the form of voids, cracks, etc. The “soft” coagulation process is achieved by increasing the affinity of the coagulant to the solvent by diluting the “hard” coagulant with the solvent or by reducing the rate of interdiffusion of the solvent from the polymer solution and the coagulant into the jet, or by increasing their viscosity.

Water, as well as ethanol, are “hard” coagulants for PANI-O solutions. The model coagulation experiments of a solution drop by these coagulators are shown in Figure 14.

Defects are formed in both cases. It is vacuoles, the rapid growth of which at the first stage determines the coagulation rate and the final macromorphology of the system. The surface of the drop is gelled at the initial moment followed by its casual rupture and the coagulant flows into the drop. The process of vacuole growth is stopped by mixing the coagulant with the solvent from the volume of the solution droplet, decreasing the activity of the coagulant which leads to the coagulation of the vacuole surfaces and its isolation. In more detail, the kinetics of this phenomenon in the case of coagulation with water and ethanol is reflected in a series of micrographs corresponding to different coagulation times (Figure 15).

As is seen, there exist two kinds of coagulant penetration into solution drop. The first one—fast infiltration through ruptures of the gel-like periphery layer leading to vacuoles formation (arrow A). The second one—inter-diffusion, or counter diffusion of a coagulant into the drop and a solvent out of the drop, obeying to fundamental diffusion laws (arrow B). The movement of the coagulation (diffusion) front from the edge to the center of the droplet along the visible dark boundary based on the series of micrographs allowed us to estimate the displacement-time dependences for both coagulants (Figure 16).

In the first stage, the coagulant quickly penetrates the drop due to convective mass transfer with the formation of vacuoles. This process significantly (~4 times) accelerates the stationary diffusion of the coagulant into the droplet because of the averaging of the concentration gradient of the coagulant during the formation and breakthrough of the gel-like shell. The growth of vacuoles ends due to dilution of coagulant with a solvent, and the stationary process of penetration of the coagulant into the drop of a solution as well as the output of the solvent from the drop occurs at the second stage at a constant rate. The beginning of this process is indicated by arrows B in Figure 15. Water penetrates the droplet and forms approximately 2 times more defects faster than ethanol at both stages.

The addition of a non-solvent to solutions has a direct effect on the diffusion fluxes of the coagulant and solvent. Data on the diffusion front rate for aqueous or ethanol solutions are shown in Figure 17.

Despite a significant increase in the viscosity of solutions structured by the presence of coagulation agents the rate of their coagulation increases compared with binary solutions. This fact could be explained by an increase in the defectiveness of the samples, but the morphological data indicate a significant decrease in defects when non-solvents are presented in PANI-O solutions (Figure 18).

As is seen from the series of micrographs shown in Figure 18, the presence of 10% water or 20% ethanol in the solution becomes the morphology of the coagulated drop practically defect-free.

The reasons for the increase in the viscosity of PANI-O solutions upon the addition of ethanol or water with a simultaneous increase in the coagulation rate of three-component solutions can be associated with the structuring mechanism. The schematic image is shown in Figure 19.

NMP solvates the polymer in a PANI-O solution, reducing the intermolecular interaction between the dianhydride and tetraamine fragments. The addition of a non-solvent capable of interacting with NMP creates conditions to associate neighboring polymer chains due to the partial “deactivation” of the solvent resulting from its hydration with water or solvation with ethanol. This process structurizes the system, which affects its rheological behavior. At the same time, the presence of a coagulant in the solution lowers the osmotic pressure during phase separation of the solution on enriched by polymer and solvent phases. This process contributes to decreasing defects, on the one hand, and reducing the portion of the coagulant required for the beginning of phase separation, on the other hand. In other words, the role of osmotic pressure and specific interactions of the solution components appear to be more important role for accelerating the defectless coagulation process, than the gradient of the coagulant content outside and inside the droplet (jet), as a traditional driving force of diffusion.

## 4. Conclusions

The conditions for the synthesis of PANI-O in an NMP solution were developed and implemented. The effect of temperature and time on the stability of the reaction solutions were analyzed, and conditions for long-term storage and processing of “living” pre-polymer solutions into fibers were determined. It was found that an increase in the molecular weight of the polymer and its concentration in solution leads to the formation of highly structured systems owing to strong intermolecular interactions.

The effect of non-solvents addition, such as water and/or ethanol, on the rheological properties of PANI-O solutions in NMP and the kinetics of their coagulation was studied. It was shown that in a certain concentration range (up to 20% for ethanol, and up to 10% for water) these non-solvents form combined solutions with PANI-O in NMP. The addition of ethanol contributes to the formation of a less strong macrostructure in the system compared to water.

The coagulation of systems with water and ethanol proceeds in two stages. At the first stage macro defects—vacuoles are formed, and at the second one, coagulation occurs due to the interdiffusion process. The presence of non-solvents in solutions leads to a significant decrease of defectiveness in the coagulated solution drops accompanying acceleration of the coagulation process. On the whole, the above observations open up new possibilities to obtain defect-free fibers from PANI-O solutions using simple, eco-friendly and available coagulants.

## Figures and Tables

**Figure 1 polymers-12-02454-f001:**
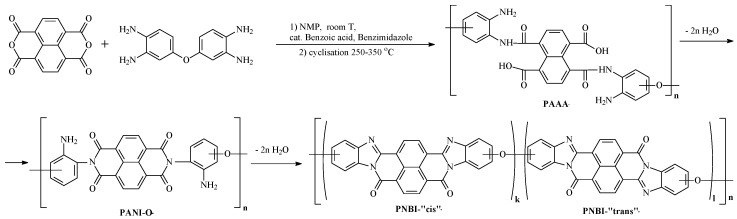
Scheme of polynaphthoylenebenzimidazole (PNBI) synthesis.

**Figure 2 polymers-12-02454-f002:**
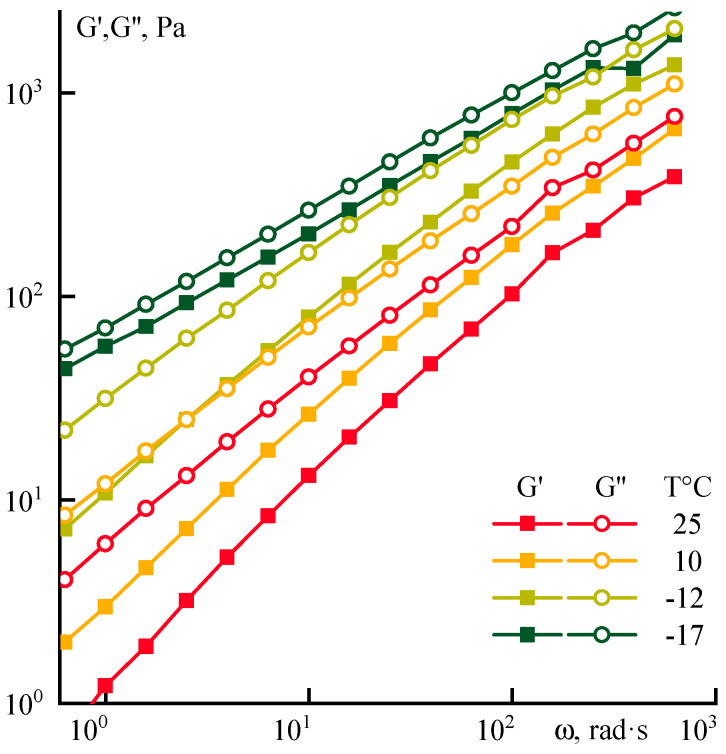
Viscoelastic properties of 8% P1 solution at different temperatures.

**Figure 3 polymers-12-02454-f003:**
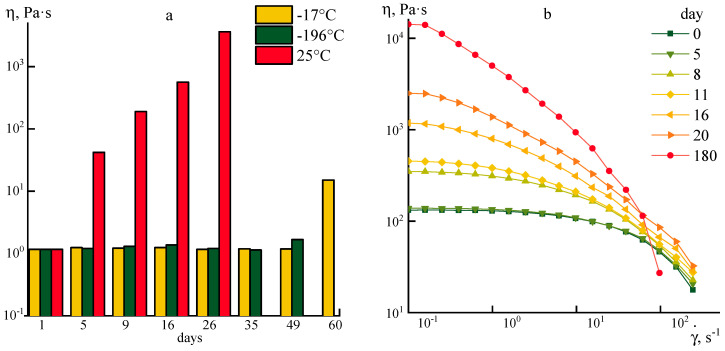
(**a**) Changes in the maximal Newtonian viscosity measured at 25 °C for P2 solution stored prolong time at different temperatures, and (**b**) flow curves of 10% P1 solutions stored at 25 °C, over time.

**Figure 4 polymers-12-02454-f004:**
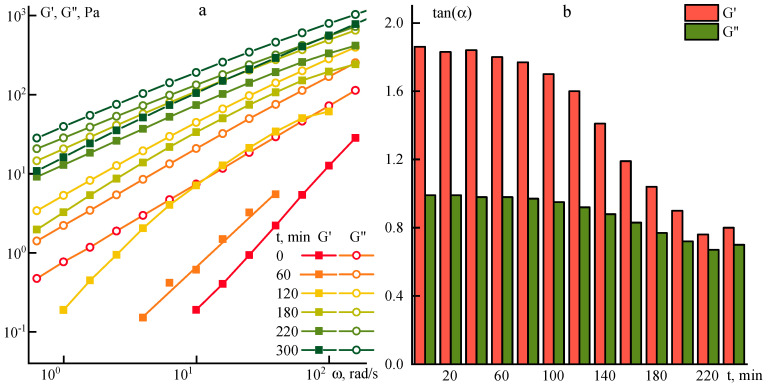
(**a**) Frequency dependences of the viscoelastic characteristics and (**b**) the kinetics of slopes change of the storage and loss moduli frequency dependences at 70 °C for a 10% P2 solution after different heating times at 70 °C.

**Figure 5 polymers-12-02454-f005:**
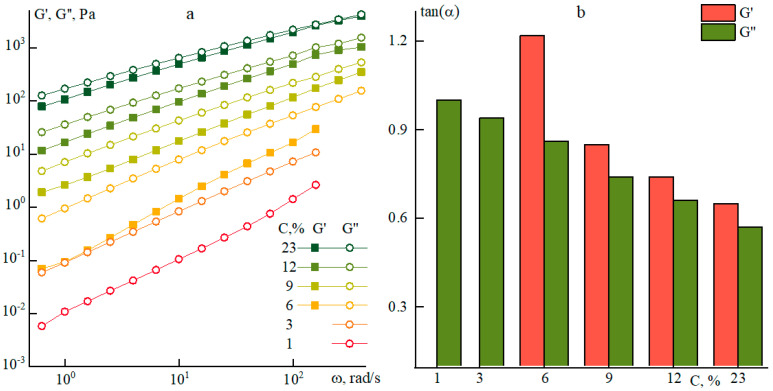
(**a**) Frequency dependences of elasticity and loss moduli and (**b**) a slope of the corresponding frequency dependences of moduli on solution P1 concentration at 25 °C.

**Figure 6 polymers-12-02454-f006:**
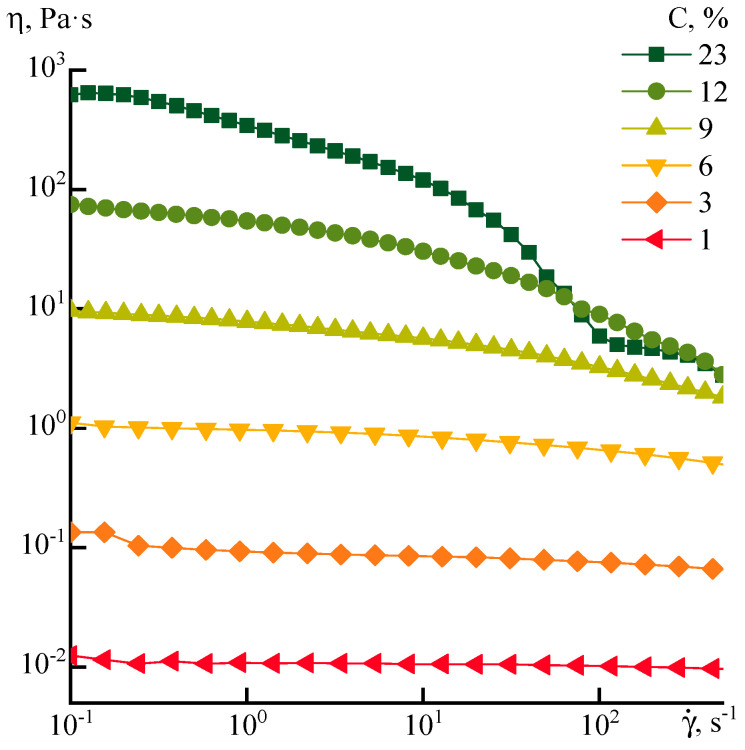
Flow curves of P1 solutions of different concentrations.

**Figure 7 polymers-12-02454-f007:**
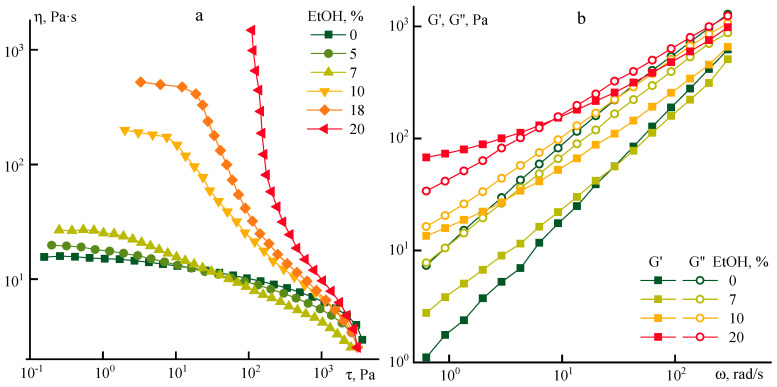
(**a**) Flow curves and (**b**) frequency dependences of moduli of prepolymer-poly-(o-aminophenylene)naphthoylenimide (PANI)-O solutions with ethanol addition.

**Figure 8 polymers-12-02454-f008:**
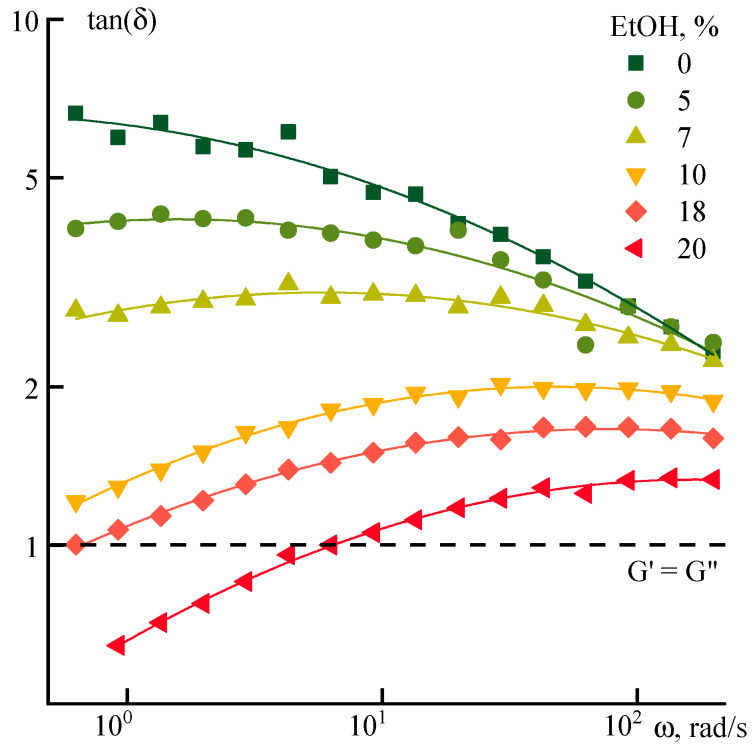
Dependence of the tan(δ) for PANI-O solutions containing different amounts of ethanol (indicated on the graph).

**Figure 9 polymers-12-02454-f009:**
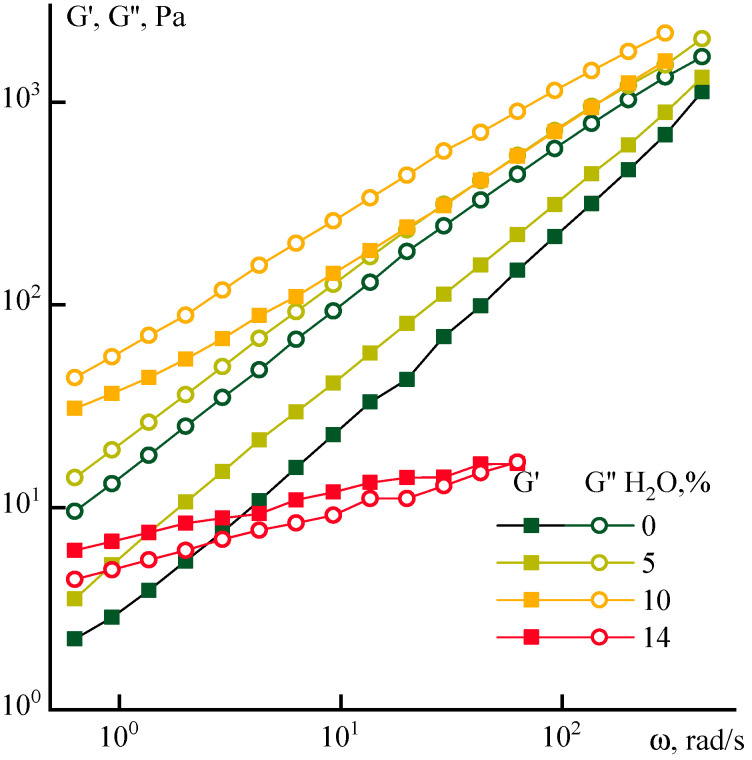
Frequency dependences of dynamic moduli of PANI-O solutions containing different content water.

**Figure 10 polymers-12-02454-f010:**
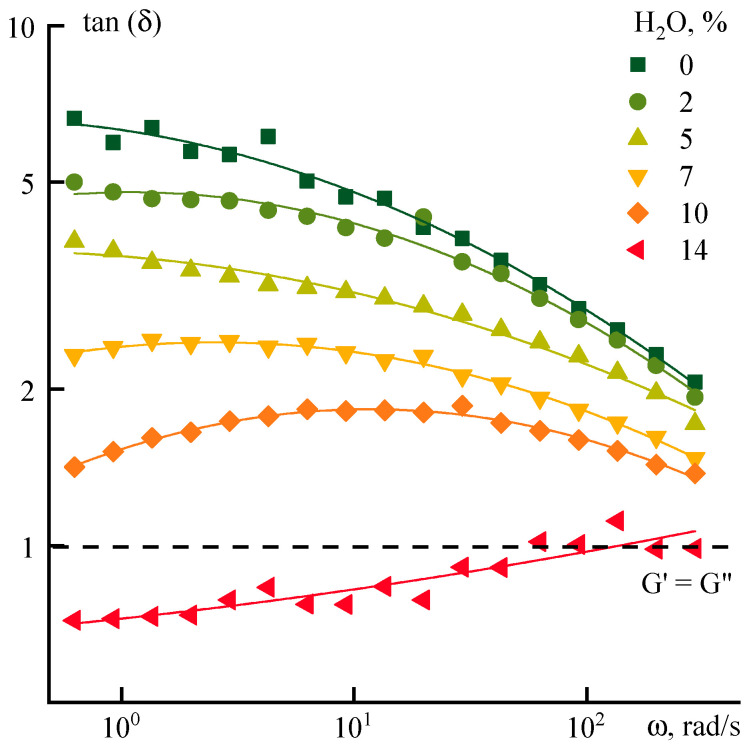
Dependence of the tan(δ) for PANI-O solutions with water addition on frequency.

**Figure 11 polymers-12-02454-f011:**
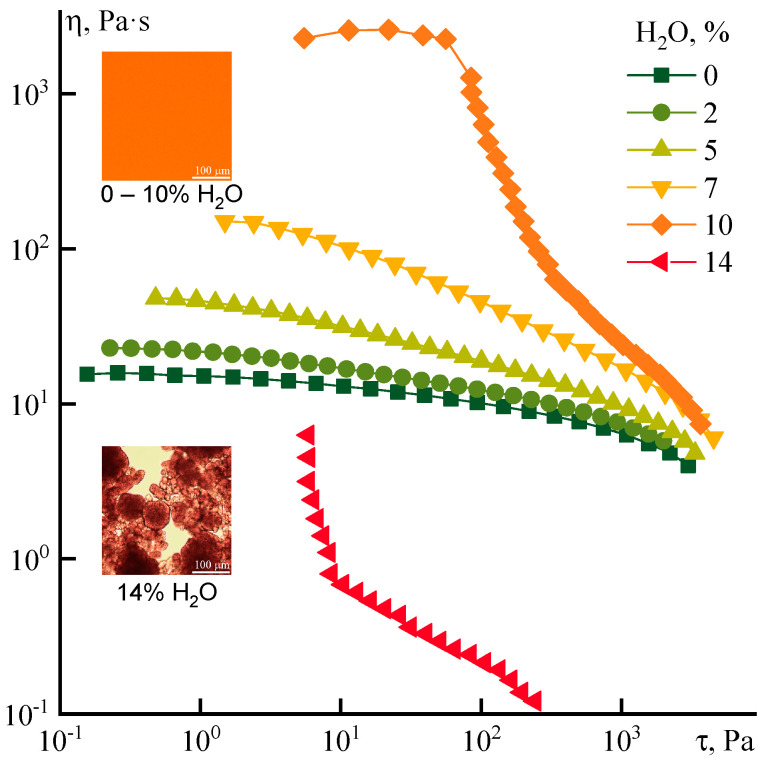
Flow curves of PANI-O solutions with different water content (indicated on the graph). The insets shows a morphology of the systems.

**Figure 12 polymers-12-02454-f012:**
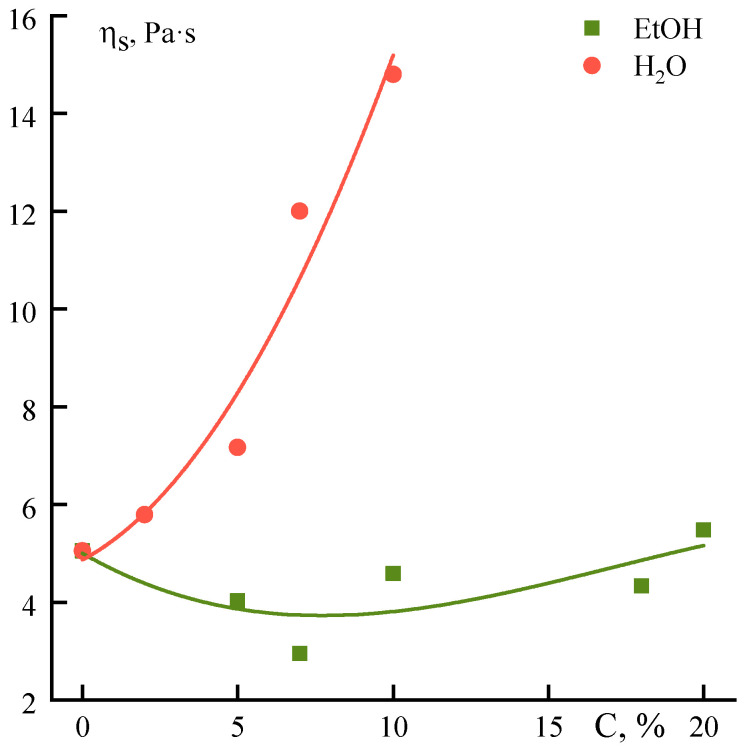
Dependence of viscosity values at the spurt beginning on the content of water and ethanol introduced into the solution.

**Figure 13 polymers-12-02454-f013:**
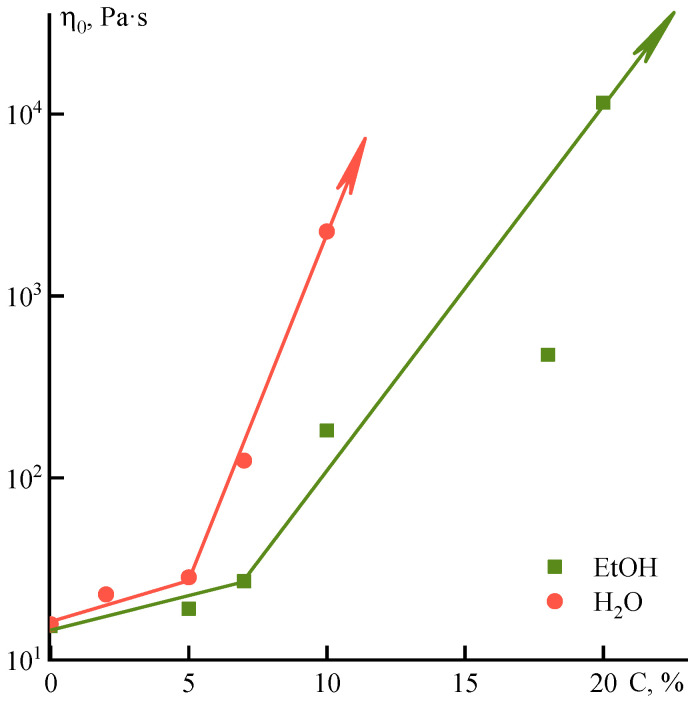
Dependence of the Newtonian viscosity on the concentration of ethanol and water introduced into the solution.

**Figure 14 polymers-12-02454-f014:**
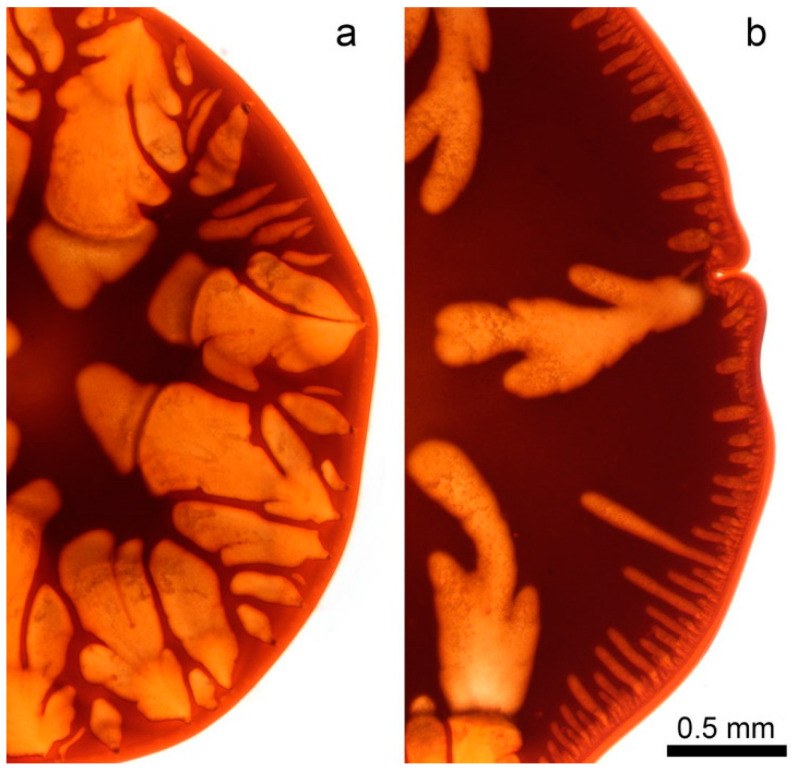
Model drop coagulation of PANI-O solution in N-methyl-2-pyrrolidone (NMP) with (**a**) water and (**b**) ethanol.

**Figure 15 polymers-12-02454-f015:**
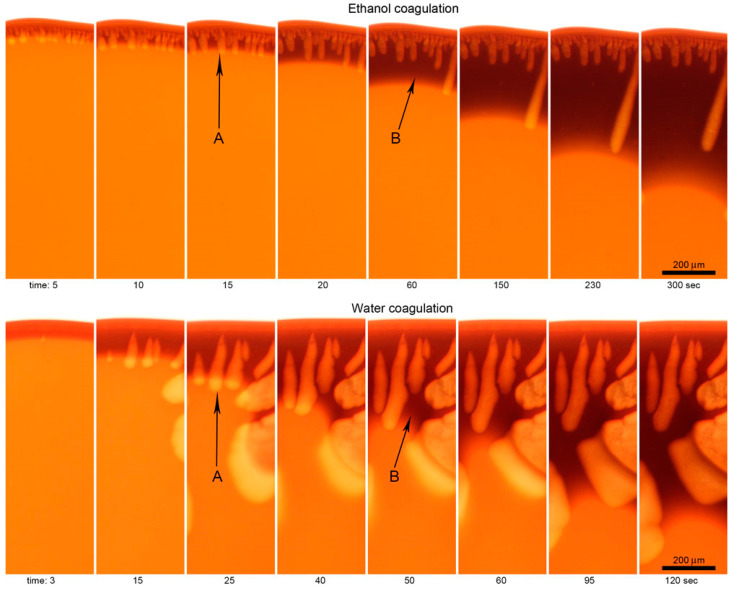
Micrographs of the selected coagulation stages of PANI-O solutions by ethanol and water. (**A**) growth of vacuoles, (**B**) diffusion coagulation.

**Figure 16 polymers-12-02454-f016:**
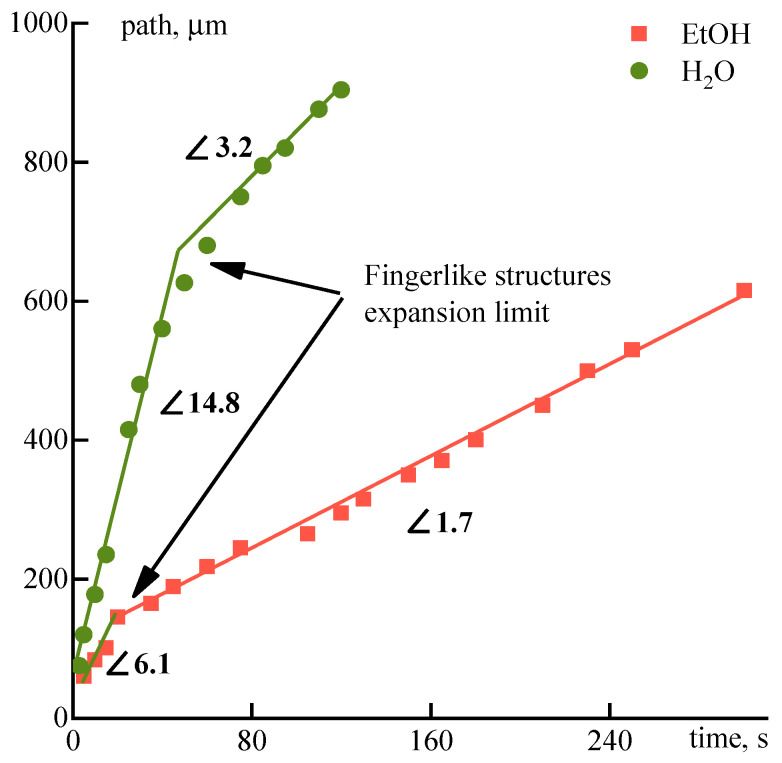
Coagulation front movement in a drop of a solution under the action of ethanol and water.

**Figure 17 polymers-12-02454-f017:**
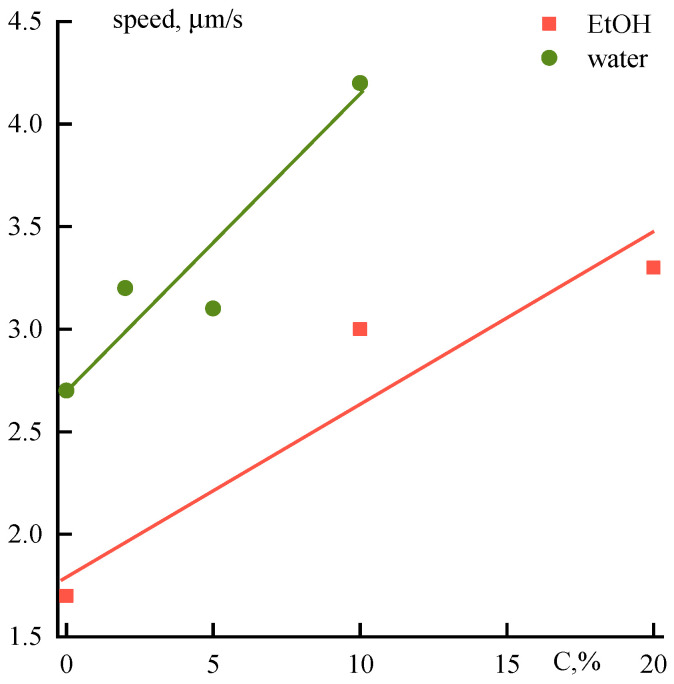
Dependence of the diffusion front rate into a drop of a solution containing alcohol or water on their content.

**Figure 18 polymers-12-02454-f018:**
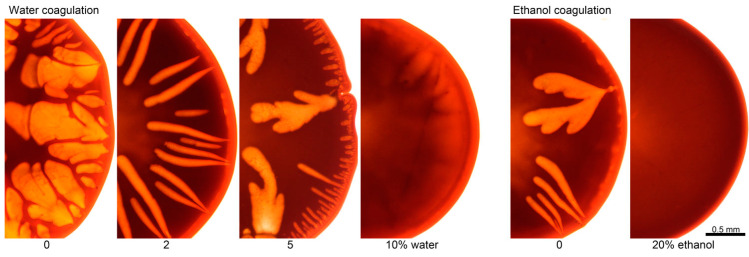
The drops morphology of PANI-O solutions containing different amounts of water and ethanol, coagulated by appropriated coagulant.

**Figure 19 polymers-12-02454-f019:**
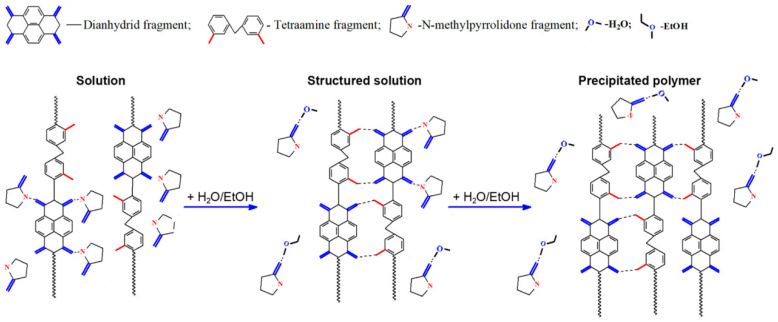
Schematic image of the interaction of PANI-O with a solvent and a non-solvent.

**Table 1 polymers-12-02454-t001:** Samples abbreviations and characteristics.

Sample	P1	P2	P3	P4	PH	PE
Concentration, wt%	1–23	10	10	10	13	13
Intrinsic viscosity, dL/g	1.5	0.6	1.2	1.8	1.2	1.2
M_υ_, kg/mole	44.2	15.8	34.4	54.3	34.4	34.4
Non-solvent, wt% to NMP	-	-	-	-	H_2_O 2, 5, 7, 10, 14	EtOH 5, 7, 10, 18, 20

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
