# Peer review of "Solubility, Rheology, and Coagulation Kinetics of Poly-(O-Aminophenylene)Naphthoylenimide Solutions"

_polymers, 2020, doi:10.3390/polym12112454_

Round 1
Reviewer 1 Report
The manuscript (polymers-978400) presents a interesting and valuable study on the coagulation of PANI-O solution at various conditions. The results have been well organized, and the conclusions are well supported. The manuscript is publishable after minor revisions according to the following comments.
- For the effects of storage temperature, could authors provide how molecule weight changes?
- The addition of Water or ethanol in PANI-O solution leads to gelation and even phase separation. Similar phenomena have been observed in other solution system, such as polyacrylonitrile. The authors state that the phase separation occurs at a water concentration of 14%, could authors provide more evidences for phase separation?
- Figure 18, ethanol coagulation. Could author provide intermediate images?
Author Response
First of all, we appreciate reviewer’s activity in attentive consideration of this manuscript. Your comments are very useful for improving the final text. Our answers on your notes see below.
- For the effects of storage temperature, could authors provide how molecule weight changes?
It was more convenient for us to estimate the change in molecular weight during solutions storage by intrinsic viscosity values. As an example of this kind of estimation, see this phrase: “To monitor the intrinsic viscosity evolution, the P2 solution was heated for 3 and 5 h, and its values became 1.2 and 1.8 dL/g, respectively. This fact indicates a significant increase in the molecular weight of the sample upon heating and a corresponding increase in the structuring of the solutions when [η] exceeds 1 dL/g” (lines 243-246). According to data of Table 1, this means that molecular weight has increased from 15.8 in reaction solution to 34.4 (3 hours storage) and 54.3 in 5 hours storage. Intensity of molecular weight growth in time increases with storage temperature.
- The addition of Water or ethanol in PANI-O solution leads to gelation and even phase separation. Similar phenomena have been observed in other solution system, such as polyacrylonitrile. The authors state that the phase separation occurs at a water concentration of 14%, could authors provide more evidences for phase separation?
When we introduce non-solvent to solution, an activity of solvent decreases, and at critical content of non-solvent a system undergoes the phase separation. According to basic thermodynamic laws, it may be either amorphous, or crystalline or liquid crystalline equilibriums. For ternary system of PANI-O/NMP/water (ethanol), the most probable is amorphous phase equilibrium with transition points forming binodal. Because of change the Flory-Huggins parameter, the system moves under binodal, that accompanies by phase separation. There are various evidences of the phase separation: optical and rheological. Optical images of solution containing 10% and 15% of water are shown in insets of Fig. 11. (Now we added they) Among rheological evidences of phase separation are the following: appearance of the yield stress (Fig. 7a), drastic increase of the storage modulus and flattening its frequency dependence (Fig. 7b), slippage effect (Figs 9, 11).
- Figure 18, ethanol coagulation. Could author provide intermediate images?
Unfortunately, we did not take photos of the coagulated drop of solutions, containing intermediate content of ethanol.

Reviewer 2 Report
In the reviewed work authors studied the effect of temperature and storage time at a constant temperature on the stability of poly-(o-aminophenylene)naphthoylenimide solutions in NMP by rotational rheometry. In the course of the work the influence of polymer concentration
and its molecular weight on the rheological properties of solutions that govern the processing behaviour of fibers was investigated. The manuscript is well-structured and logically arranged, with proper desrciption of the results obtained.
However, some points need more attention:
- "PANI" abbreviation is commonly used for polyaniline;
- In this research authors performed the experiments in the temperature range of -17 – +70°C - what is the temperature effect above 70 deg C as some processing techniques may require higher temperatures;
- Considering the interactions of PANI-O with a solvent and a non-solvent - are the structures showed in Fig. 19 speculative or there are any spectroscopic evidences for their formation? What is the temperature effect on structured solution?
- Minor mistakes: e.g. Line 167 "Kinetics measurements", 180: "PANI-O appears to be capable of chemical and physicochemical changes in a solution ... (style), etc.
- In Tab. 1 authors give Mv as molar mass [kg/mol], but in sub-section 3.1.1 it is dimensionless molecular weight -please unify,
- Authors state (Lines 284-286) that "transformation of inter- and intramolecular interactions in solutions in a presence of non-solvent could create new rheological behavior, as well as the specific thermodynamic and kinetic conditions at interaction with a coagulant" - please explain and expand this thought.
Author Response
Authors express many thanks to reviewer for detailed analysis of the manuscript and important notes:
- “PANI” abbreviation is commonly used for polyaniline;
Yes, the reviewer is right, but we tried to repeat titles of chemical fragments of polymers under investigation: Poly-(o-Aminophenylene)NaphthoylenImide, but with adding “O” we point to oxygen in the main chain. Also, that discerns this polymer from polyaniline.
- In this research authors performed the experiments in the temperature range of -17 – +70°C - what is the temperature effect above 70 deg C as some processing techniques may require higher temperatures;
The rate of intramolecular cyclization and polymerization in solution accelerates exponentially at heating. The solution remains stable for about an hour at 70°C. At a higher temperature, the reactions proceed even faster and, in addition, are accompanied by the evaporation of the active solvent. That is why, it is not reasonable to process such solutions at higher temperatures.
- Considering the interactions of PANI-O with a solvent and a non-solvent - are the structures showed in Fig. 19 speculative or there are any spectroscopic evidences for their formation? What is the temperature effect on structured solution?
Fig. 19 shows only the possible scheme of the polymer-solvent-non-solvent interactions that we assume. Of course, we are planning to perform the IR spectroscopy tests of solutions.
The effect of temperature on the structuring of solutions has not been studied carefully, but based on common regularities of chemical processes, we can expect increase of chemical transformations in polymer, and simultaneously decrease of the entanglement network density. As reviewer could understand, this manuscript is a base for novel fibers spinning, and further we will draw additional attention to time-temperature regimes of these solutions processing.
- Minor mistakes: e.g. Line 167 "Kinetics measurements", 180: "PANI-O appears to be capable of chemical and physicochemical changes in a solution ... (style), etc.
We reviewed the document and tried to correct all stylistic errors.
- In Tab. 1 authors give Mv as molar mass [kg/mol], but in sub-section 3.1.1 it is dimensionless molecular weight -please unify,
Table 1 provides calculated molecular weights using Mark-Kuhn-Houwink equation. According this approach, the intrinsic viscosity is a measure of molecular weight. The dimension of intrinsic viscosity is dL/g.
- Authors state (Lines 284-286) that "transformation of inter- and intramolecular interactions in solutions in a presence of non-solvent could create new rheological behavior, as well as the specific thermodynamic and kinetic conditions at interaction with a coagulant" - please explain and expand this thought.
According the text below these lines, we show that in presence of such non-solvent as water or ethanol the solution viscosity and dynamic moduli increase allowing us to use controlled storage time-temperature to reach necessary viscoelastic properties of solutions for successful processing in fibers. In addition, introducing definite amounts of coagulant into dopes leads to almost defectless morphology of coagulated drop as a model of the fiber cross section.
